# Exposing VLM Memorization of Famous Landmarks: A 55K Building Age Dataset Revealing Popularity Bias

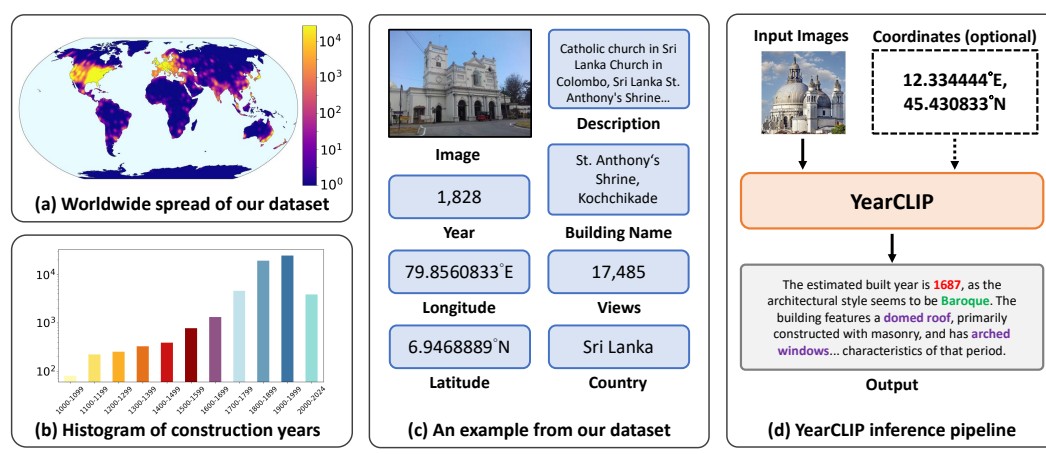

Figure 1: **Overview of YEARGUESSR and YearCLIP.** (a) Global distribution of the 55k Wikipedia-sourced building images. (b) Log-scale histogram of construction years spanning 1001~2024 CE. (c) Display all the attributes with an example in **YEARGUESSR** (d) Given an image and optional GPS coordinates, **YearCLIP** returns the estimated construction year together with an architectural rationale.

## Abstract

Building age is a crucial yet underexplored factor for sustainability, heritage, and safety, lacking a public benchmark that is both global and ordinal. We uncover that state-of-the-art vision-language models achieve up to 34% better accuracy on famous landmarks compared to ordinary buildings, suggesting they memorize popular structures from training data rather than learning generalizable architectural features. To investigate this phenomenon, we introduce the largest open benchmark for building age estimation: the **YearGuessr** dataset and our proposed baseline model, **YearCLIP**. **YearGuessr** comprises 55,546 Wikipedia facades from 157 countries, despite geographic skew toward Western architecture, with continuous ordinal labels from 1001 to 2024 CE and rich multi-modal attributes including GPS, captions, and page-view counts. We frame age prediction as ordinal regression and introduce popularity-based MAE plus interval accuracy ($\pm$ 5/20/50/100 yr). In addition, we benchmark 30+ models, including CNN-based, Transformer-based, CLIP-based models, and VLMs. Our **YearCLIP** model shows ordinal training halves MAE, while GPS priors further reduce the error. Zero-shot VLMs excel on landmarks but struggle on unrecognized buildings, exposing a popularity bias that our metric captures. We will make our dataset and code publicly available and offer the largest open benchmark for building age estimation and reasoning.

Table 1: **Building-age datasets.** Earlier datasets are either regional, post-1900, have no images, or are closed source. Our YEARGUESSR will be the first CC BY-SA 4.0 image set with continuous labels, global (157 countries) coverage, and a 1001~2024 CE span.

| Dataset | Region(s) | Year span | Size | Modalities | Continuous labels | Images | Open? |
|---|---|---|---|---|---|---|---|
| MyCD (Dionelis et al., 2025) | Europe | $<1930 \sim >2006$ | 32k img | SVI, VHR, MSI | | ✓ | ✓ |
| CMAB (Zhang et al., 2024b) | China | $1985 \sim 2018$ | 29M bldg | RS, SVI, GIS, HS | ✓ | | ✓ |
| MTBF-33 (Uhl & Leyk, 2022) | USA (33 counties) | $1900 \sim 2015$ | 6.2M img | Footprints (SHP) | ✓ | ✓ | ✓ |
| ResBldgAge (Rosser et al., 2019) | UK | $<1915 \sim >1980$ | 2.5k img | Maps, census | ✓ | | |
| 3D-GIS Age (Biljecki & Sindram, 2017) | NL | $1860 \sim 2017$ | 35k bldg | LOD1 3-D GIS | ✓ | | |
| UrbanFormAge (Nachtigall et al., 2023) | NL, FR, ES | $<1945 \sim 2010$ | 25.3M img | OSM, 2-D urban form | ✓ | ✓ | ✓ |
| GPT-4V London (Zeng et al., 2024) | London | $<1700 \sim >2020$ | 131 img | Facade + attrs | | ✓ | ✓ |
| PhotoAge (Zeppelzauer et al., 2018) | Austria | $1960s \sim 2010s$ | 11.1k img | real estate, web | ✓ | ✓ | |
| StreetViewAge (Sun et al., 2021) | NL (AMS) | $1300 \sim 2000$ | 39k img | SVI + BAG | | ✓ | ✓ |
| WikiChurches (Barz & Denzler, 2021) | Europe | $401 \sim 2011$ | 9.5k img | Wikipedia facades | ✓ | ✓ | ✓ |
| **YEARGUESSR (ours)** | 157 countries | $1001 \sim 2024$ | 55.5k img | Wikipedia facades | ✓ | ✓ | ✓ |

# 1 INTRODUCTION

The construction year of a building is essential for sustainability audits, heritage preservation, and post-disaster assessment. Yet, unlike style or material, the age of most of the world's 1.5 trillion mapped buildings is unknown. Automated *building age estimation* could support continent-scale retrofitting and fine-grained historical queries, but progress is hindered by the lack of a **global, large-scale, open benchmark**. Figure 1 previews **YearGuessr**, our answer to this gap.

As shown in Table 1, existing datasets are either geographically narrow or temporally shallow. *MyCD* dataset (Dionelis et al., 2025) covers only Western Europe with coarse epochs (<1930, 1930~1959, ...), *CMAB* (Zhang et al., 2024b) is restricted to modern Chinese facades and narrow temporal range (1985~2018), *MTBF-33* (Zeng et al., 2024) lacks photographs, and others remain geographically or temporally narrow. Prior works also cast age prediction as classification, ignoring temporal ordinality, while licensing restrictions impede reproducibility.

Beyond data gaps, a fundamental question remains: ***do vision-language models (VLMs) truly learn architecture, or simply memorize landmarks?*** We show that VLMs like Gemini-2.0 gain +34.18% accuracy on high-popularity buildings, whereas other general models degrade (Section 5). It depicts the evidence of *popularity memorization* rather than genuine architectural understanding.

To address these issues, we present **YearGuessr**, the first open benchmark for building-age estimation with **55,546 images across 157 countries, spanning 1001–2024 CE** in Figure 1(a) and (b). Each entry includes facades, GPS, captions, and Wikipedia page views to probe memorization bias. We frame the task as ordinal regression and evaluate with MAE, Interval Accuracy, and a new popularity-aware metric, demonstrating that VLMs' performance might stem from memorization rather than architectural understanding potentially.

Our contributions are:

- **YearGuessr dataset**: will be the first open, CC BY-SA 4.0 corpus with global coverage, large-scale ordinal labels, GPS, and rich textual descriptions.
- **Evaluation protocol**: an original regression benchmark with Interval Accuracy and a new popularity-based MAE metric to quantify memorization bias.
- **Baselines**: a study of 30+ CNN-, Transformer-, CLIP-based, and LLM/VLM models.
- **Insights**: We reveal that VLMs memorize popular landmarks, achieving dramatically different performance based on building fame rather than architectural features.

# 2 RELATED WORK

**Datasets for building-age estimation.** Prior corpora remain limited in scope. Building Age Estimation (Dionelis et al., 2025) fuses multiple modalities but covers only Western Europe and coarse epochs. CMAB adds Chinese façades yet spans 1985–2018 (Zhang et al., 2024b). MTBF-33 lists U.S. footprints without imagery (Uhl & Leyk, 2022). Other efforts are small-scale (131 London photos (Zeng et al., 2024)) or style-focused (WikiChurches (Barz & Denzler, 2021)). Early work on architectural style recognition (Mathias et al., 2012; 2016; Martinović et al., 2012) and heritage analysis (Llamas et al., 2017) established foundations for visual classification. Broader

surveys (Fiorucci et al., 2020) situate AI in preservation. Earlier datasets often rely on real-estate photos or GIS metadata (Sun et al., 2021; Li et al., 2018; Biljecki & Sindram, 2017). In contrast, YEARGUESSR contributes 55k Wikipedia-sourced CC-BY-SA images spanning 1001–2024 CE across six continents, following best practices from WIT (Srinivasan et al., 2021).

**Image geolocalization foundations.** Planet-scale localization began with Im2GPS (Hays & Efros, 2008) and PlaNet (Weyand et al., 2016), later refined via hierarchical cells (Vo et al., 2017; Seo et al., 2018), scene cues (Muller-Budack et al., 2018), and segmentation (Pramanick et al., 2022). Recent work integrates transformers (Clark et al., 2023), cross-view supervision (Zhu et al., 2022; Bastani et al., 2023), and human traces (Luo et al., 2022). Related building-attribute studies predict materials, structure, or energy efficiency (Bell et al., 2015; Dimitrov & Golparvar-Fard, 2014; Seyedzadeh et al., 2018). Advanced façade parsing (Liu et al., 2020; Zhang et al., 2022a) highlights age-indicative elements. Despite progress, these pipelines stop short of fine-grained building-age dating.

**Geo-aware VLMs.** Contrastive VLMs lack strong spatial priors. GeoCLIP (Vivanco Cepeda et al., 2023), LLMGeo (Wang et al., 2024), GeoReasoner (Li et al., 2024), and PIGEON (Haas et al., 2024) incorporate coordinates or climate. SNAP (Sarlin et al., 2023) uses map tiles, while AddressCLIP (Xu et al., 2024) aligns directly with street addresses. Metadata-image fusion shows promise (e.g., EXIF-as-language (Zheng et al., 2023)), paralleling our approach. Yet our evaluation also reveals popularity bias (Zhao et al., 2020), echoing dataset bias studies (Wang et al., 2022; Shankar et al., 2017).

**Ordinal regression with numeric cues.** Ordinal regression evolved from CORAL/CORN losses (Niu et al., 2016; Cao et al., 2020) to order-regularised and probabilistic methods (Guo et al., 2021; Li et al., 2021; Shin et al., 2022). Classification with discretization (Fu et al., 2018) proved effective for continuous estimation. Vision-language extensions (OrdinalCLIP (Li et al., 2022), NumCLIP (Du et al., 2024)) encode order in text. Related applications include age (Rothe et al., 2015), depth (Zhang et al., 2022b), and counts (Liang et al., 2023). We extend this line by evaluating CLIP and GPT-4V (Achiam et al., 2023) on large-scale building ages.

**Multi-modal learning signals.** Auxiliary cues such as weather, land cover, or captions (Haas et al., 2024; Yang et al., 2021; Radford et al., 2021) aid localization. WikiTiLo (Zhang et al., 2024a) shows VLMs still lack temporal–geo knowledge, motivating visual chain-of-thought prompting (Chen et al., 2024b). Bias mitigation strategies (Wang et al., 2020) and diversity concerns (Shankar et al., 2017) remain critical. Inspired by NeRF's RFFs (Mildenhall et al., 2021), we combine coordinate encodings with ordinal losses to achieve state-of-the-art on YearGuessr.

## 3 DATASET AND BENCHMARK

### 3.1 DATASET CONSTRUCTION

**Data sources and licences..** All images and textual descriptions are scraped from *Wikipedia* and its sister project *Wikimedia Commons*. Both platforms distribute user-contributed content primarily under the CC BY-SA 4.0 or Public-Domain licences,[1] which permit redistribution provided attribution is preserved. During crawling, we query the Commons API for the exact licence of every file and discard items tagged as *"non-free"* or *"no derivatives"*. Consequently, the **YearGuessr** corpus (images, metadata, and split indices) will be publicly available under the same CC BY-SA 4.0. The accompanying code will be publicly available under the MIT license.

**Automatic collection pipeline.** Figure 2 (a) illustrates the four-stage crawler. (1) We begin by recursively traversing the `Buildings_and_structures_by_year_of_completion` category on Wikimedia Commons, dated from 1001 CE to 2024. (2) It collects 90,230 pages related to buildings and structures from the category. (3) For every building page, we extract the first infobox image, geographic coordinates, and full wikitext. (4) We call the Wikimedia Pageviews API to obtain the total number of views between 01 Jul 2023 and 01 Jul 2024, which we later use as a proxy for *popularity*.

**Cleaning and quality control.** We remove duplicates, off-topic images, and irrelevant files via the Figure 2 pipeline. First, deduplication retains one image per page title, eliminating 8,346 duplicates. Next, a ViT-B/32 CLIP filter scores similarity to "a building facade" and drops 26,338 low matches.

---

[1]https://en.wikipedia.org/wiki/Wikipedia:Copyrights

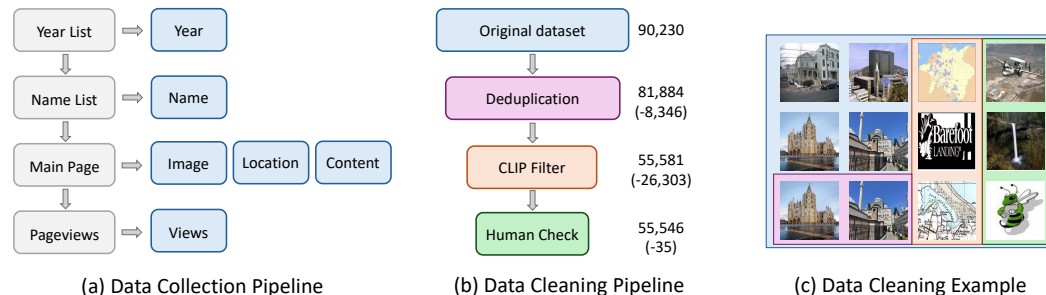

(a) Data Collection Pipeline  (b) Data Cleaning Pipeline  (c) Data Cleaning Example

Figure 2: **Data collection and cleaning pipeline.** (a) We crawl the Wikipedia category tree of buildings, collecting façade images, construction years, GPS coordinates, textual descriptions, and pageview statistics. (b) The raw crawl of 90k images is refined through deduplication, a CLIP-based building filter, and a light manual audit, yielding 55k clean façades. (c) Examples of discarded non-building or duplicate samples.

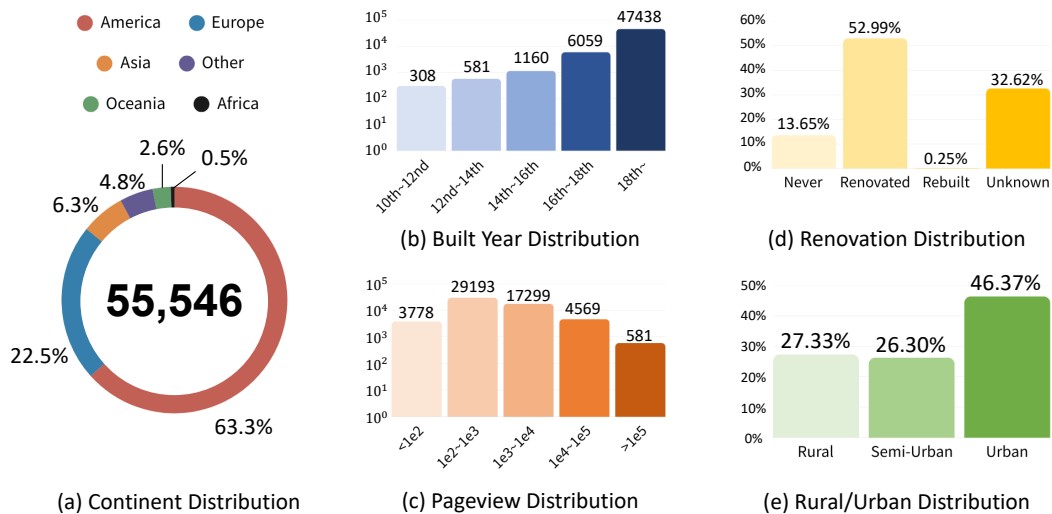

(a) Continent Distribution  (c) Pageview Distribution  (e) Rural/Urban Distribution

Figure 3: **Dataset statistics.** This figure provides an overview of our dataset's key characteristics. (a) Continent Distribution shows the geographical origins of the images. (b) Built Year Distribution illustrates the age of the structures. (c) Pageview Distribution represents the buildings' popularity. (d) Renovation Distribution indicates the extent of reconstruction. (e) Rural/Urban Distribution reflects the population density of a building's location.

Finally, a brief manual audit of the test split removes 35 obvious outliers (e.g., aircraft, interiors). This leaves 55,546 unique, high-quality facade images, each representing a distinct building.

**Train/validation/test split.** We stratify by *construction decade* and *continent* and then assign 60% / 20% / 20% of buildings to the training, validation and test partitions, respectively, resulting in 33,337 / 11,122 / 11,087 samples. No building, caption, or image appears in more than one split.

**Metadata completeness.** All samples include GPS coordinates (100%), country name (via reverse-geocoding), and textual description (median length 2,240 characters). Figure 1 (b) shows the year distribution spanning 1,000 years. Figure 1 (a) shows geographical spread across 157 countries.

### 3.2 STATISTICS ANALYSIS

**Geographic Distribution.** Our dataset spans 157 nations, but it is heavily skewed toward the Americas (63.3%) and Europe (22.5%). Asia accounts for 6.3%, while Oceania and Africa make up a smaller fraction. This geographical imbalance, visualized in Figure 3(a), motivates us to report continent-specific evaluation metrics in later sections.

**Temporal Distribution.** Figure 3(b) illustrates the age of the structures in our dataset. There's a notable concentration of buildings constructed in the 18th century and later. The dataset also includes a significant number of pre-1600 buildings, which enables large-scale historical analysis.

**Building Popularity.** We use a building's Wikipedia pageviews as a proxy for its popularity. Figure 3(c) shows a heavy-tailed distribution: a large number of images have low annual views, while a small subset of highly viewed landmarks (e.g., over 10,000 views) represent a significant portion of the corpus. This skew is a key characteristic of our dataset.

**Renovation Scenario.** Figure 3(d) summarizes building reconstruction status. Most (52.99%) show no renovation, while 32.62% lack historical records. We distinguish between *renovated* buildings, where the original construction year remains valid, and *rebuilt* buildings, where the construction year is effectively redefined. Annotations were extracted via LLM analysis of building descriptions.

**Rural/Urban Regions.** Figure 3(e) reports location categories derived by mapping coordinates to GPWv4.11 population density. Buildings are classified as *Rural* ($<300$ people/km$^2$), *Semi-urban* ($300$–$1500$ people/km$^2$), or *Urban* ($>1500$ people/km$^2$), providing finer-grained geographical context.

## 3.3 TASKS & METRICS

**Problem formulation.** Given a facade image $I$ and optional GPS coordinates $g = (\phi, \lambda)$, the primary task is to predict *construction year* $y \in \mathbb{Z}$ of the building depicted. Following Niu et al. (2016) and Cao et al. (2020), we cast the problem as **ordinal regression** rather than hard classification or naïve regression. Formally, a model $f_\theta$ outputs a scalar $\hat{y} = f_\theta(I, g)$ and may additionally emit a text rationale $\hat{r}$ (Section 5).

**Evaluation metrics.**

(1) Mean Absolute Error: MAE $= \frac{1}{N} \sum_{i=1}^{N} |y_i - \hat{y}_i|$.

(2) Interval Accuracy: $\text{IA}_k = \frac{1}{N} \sum_{i=1}^{N} \mathbb{1}[|y_i - \hat{y}_i| \leq k]$ for $k \in \{5, 20, 50, 100\}$ years. $\text{IA}_{20}$ approximates "gets the right architectural period"; $\text{IA}_5$ rewards near exact dating.

**Evaluation protocol.** All metrics are computed in the fixed test split (11,087 images as mentioned in Section 3.1). For models that consume GPS, both *with-* and *without*-location scores are reported to expose the benefit and potential leakage of spatial priors. We repeat every experiment with three random seeds and report the mean and standard deviation.

# 4 OUR YEARCLIP MODEL

**Model architecture.** Traditional models for image-based tasks often rely on CNNs or Transformers. Instead, we use CLIP (Radford et al., 2021), a multi-modal framework pre-trained to align image and text, ideal for the semantically rich task of building age prediction due to its zero-shot generalization to under-represented periods. To combine CLIP with ordinal regression, we adopt NumCLIP (Du et al., 2024), which uses language priors for a coarse-to-fine strategy. It first classifies architectural styles coarsely, then computes similarity scores for a regressor to predict fine-grained years, balancing style recognition with temporal precision. The overall pipeline design is shown as Figure 4.

**Location conditioning.** The GeoReasoner framework (Li et al., 2024) indicates that building images degrade performance in geographic location prediction, likely due to colonial influences or style imitation across regions. Geographic context, however, provides cues absent in visual data, enhancing architectural style interpretation for age estimation. We utilize the pretrained location encoder from GeoCLIP (Vivanco Cepeda et al., 2023) and fuse location and image embeddings in the latent space to integrate spatial information. For images lacking location data, which is a common scenario, we rely solely on image embeddings for similarity computation, ensuring model flexibility.

**Zero convolution.** Fusing image and location embeddings via a weighting parameter $\alpha$ is straightforward but challenging to optimize manually. Instead, we add a zero convolution layer post-location encoder, enabling the model to learn optimal weights autonomously during training, improving fusion effectiveness.

**Reasoning prompt integration.** Unlike the original NumCLIP, which relies solely on category similarity for regression, we augment the regressor with pre-defined reasoning prompts. These

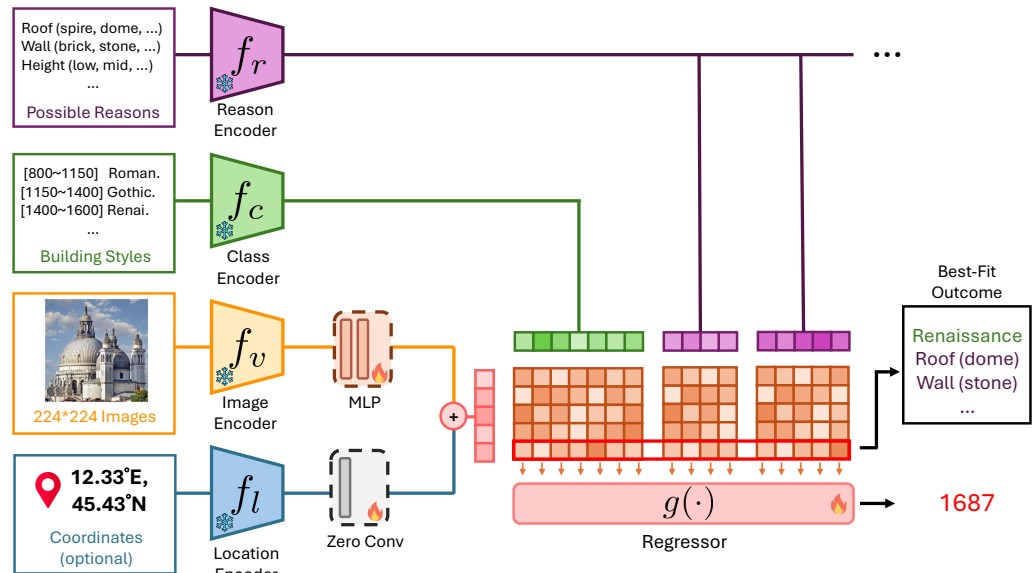

Figure 4: **YEARCLIP architecture.** An image encoder $f_v$ (CLIP) extracts 224×224 facade features. We then fuse the feature with a GPS embedding from the location encoder $f_l$ (RFF + MLP, optional input) via a learnable zero-convolution. Parallel text branches encode (i) seven coarse style classes $f_c$ and (ii) a bank of reasoning prompts $f_r$ describing roofs, walls, heights, etc. All frozen encoders feed a trainable regressor $g(\cdot)$ that performs coarse-to-fine ordinal regression. It predicts a construction year (here 1687), selects the best-matching style/reason tokens, and outputs a readable rationale.

prompts, encoded via a text encoder, enrich the input feature space, enabling more accurate year predictions. Additionally, the regressor can backtrack the importance of each input, providing insights into the model's decision-making process. This approach facilitates reasoning analysis without requiring external captioning models or vision-language models.

Due to the space limit, we provide model training settings in Appendix A.

## 5 RESULTS AND ANALYSIS

### 5.1 MAIN RESULTS

Table 2 reports 23 representative models on the YearGuessr test split (11,087 images), including CNN-based, Transformer-based, CLIP-based, Closed VLMs, and Open VLMs. Metrics include MAE, Interval Accuracy within ±5/100 years ($IA_5$, $IA_{100}$), and popularity-stratified $IA_5$ ($<10^2$, $>10^5$) with *Gain* defined as the difference between high- and low-popularity bins.

**Ordinal regression improves fine-grained prediction.** Our YearCLIP reduces MAE to 39.52, outperforming ConvNeXt-B (44.42) and Swin-B (47.65). Compared to GeoCLIP, which does not use ordinal regression, YearCLIP decreases MAE by 13.5% ($45.69 \rightarrow 39.52$) while maintaining competitive $IA_5$ and $IA_{100}$.

**CLIP priors.** Zero-shot CLIP achieves MAE 78.23, better than some open-source VLMs like LLaVA-v16-13B (194.07) and MiniCPM-V2-6B (106.41). Fine-tuned CLIP variants (GeoCLIP, NumCLIP, YearCLIP) consistently improve MAE and IA over CNN and Transformer baselines, highlighting the benefits of pre-training and fine-tuning.

**Closed-source VLMs dominate.** Top performers are Gemini1.5-Pro (MAE 33.08), Gemini2.0-Flash (MAE 33.91), and Grok2 (MAE 35.28). The best open-source model, Gemma3-27B, scores MAE 36.48. Other open-source models like LLaVA-v16-13B and InternVL2-26B lag considerably.

Table 2: **Performance on basic metric, interval accuracy, and popularity analysis (simplified).** Please refer to Appendix C for a complete evaluation of all 43 methods.

| Method | Model | Basic | Interval Accuracy | | Popularity ($IA_5$) | | |
|---|---|---|---|---|---|---|---|
| | | MAE ($\downarrow$) | $\leq 5$ ($\uparrow$) | $\leq 100$ ($\uparrow$) | $< 10^2$ ($\uparrow$) | $> 10^5$ ($\uparrow$) | Gain |
| CNN | ResNet-50 (He et al., 2016) | $54.14 \pm 0.47$ | $10.44 \pm 0.56$ | $88.68 \pm 0.41$ | $12.39 \pm 1.38$ | $9.14 \pm 0.51$ | $-3.25 \pm 1.60$ |
| | ConvNeXt-B (Liu et al., 2022) | $\mathbf{44.42 \pm 0.33}$ | $\mathbf{14.01 \pm 0.56}$ | $\mathbf{90.72 \pm 0.07}$ | $\mathbf{16.57 \pm 0.94}$ | $\mathbf{12.68 \pm 3.11}$ | $-3.89 \pm 3.58$ |
| Transformer | ViT-B/16 (Dosovitskiy et al., 2020) | $49.16 \pm 0.43$ | $12.50 \pm 0.51$ | $89.52 \pm 0.42$ | $15.82 \pm 0.85$ | $6.78 \pm 1.84$ | $-9.04 \pm 1.75$ |
| | Swin-B (Liu et al., 2021) | $\mathbf{47.65 \pm 0.67}$ | $\mathbf{12.65 \pm 0.68}$ | $\mathbf{89.95 \pm 0.19}$ | $\mathbf{15.92 \pm 2.27}$ | $\mathbf{9.14 \pm 0.51}$ | $-6.77 \pm 1.88$ |
| CLIP-based | CLIP (zero-shot) (Radford et al., 2021) | $78.23 \pm 0.00$ | $12.78 \pm 0.00$ | $78.55 \pm 0.00$ | $13.52 \pm 0.00$ | $7.96 \pm 0.00$ | $-5.56 \pm 0.00$ |
| | GeoCLIP (Vivanco Cepeda et al., 2023) | $45.69 \pm 0.49$ | $\mathbf{23.79 \pm 0.26}$ | $89.54 \pm 0.11$ | $\mathbf{24.37 \pm 1.06}$ | $\mathbf{19.17 \pm 1.35}$ | $-5.19 \pm 0.91$ |
| | NumCLIP (Du et al., 2024) | $40.01 \pm 0.55$ | $18.15 \pm 0.31$ | $\mathbf{91.76 \pm 0.16}$ | $21.69 \pm 1.62$ | $11.80 \pm 3.58$ | $-9.89 \pm 2.42$ |
| | YearCLIP (ours) | $\mathbf{39.52 \pm 0.27}$ | $18.93 \pm 0.75$ | $91.63 \pm 0.44$ | $20.19 \pm 0.94$ | $12.39 \pm 3.86$ | $-7.80 \pm 4.26$ |
| Closed VLMs | GPT4o-mini (Achiam et al., 2023) | $42.69 \pm 0.00$ | $22.75 \pm 0.00$ | $89.62 \pm 0.00$ | $19.01 \pm 0.00$ | $48.67 \pm 0.00$ | $29.66 \pm 0.00$ |
| | Gemini1.5-pro (Gemini Team et al., 2024) | $\mathbf{33.08 \pm 0.00}$ | $28.18 \pm 0.00$ | $\mathbf{93.14 \pm 0.00}$ | $\mathbf{26.76 \pm 0.00}$ | $43.36 \pm 0.00$ | $16.60 \pm 0.00$ |
| | Gemini2.0-flash (Gemini Team et al., 2025) | $33.91 \pm 0.00$ | $\mathbf{29.71 \pm 0.00}$ | $92.75 \pm 0.00$ | $24.23 \pm 0.00$ | $\mathbf{58.41 \pm 0.00}$ | $34.18 \pm 0.00$ |
| | Claude3-haiku (Anthropic Team, 2024) | $47.88 \pm 0.00$ | $16.13 \pm 0.00$ | $88.47 \pm 0.00$ | $18.45 \pm 0.00$ | $32.74 \pm 0.00$ | $14.29 \pm 0.00$ |
| | Grok2 (xAI Team, 2024) | $35.28 \pm 0.00$ | $27.57 \pm 0.00$ | $93.02 \pm 0.00$ | $25.77 \pm 0.00$ | $42.48 \pm 0.00$ | $16.71 \pm 0.00$ |
| OpenVLM | CogVLM2-19B (Hong et al., 2024) | $42.50 \pm 0.52$ | $18.63 \pm 0.30$ | $90.57 \pm 0.18$ | $18.31 \pm 1.25$ | $23.01 \pm 1.77$ | $4.70 \pm 2.99$ |
| | Gemma3-27B (Team, 2025) | $\mathbf{36.48 \pm 0.00}$ | $\mathbf{25.58 \pm 0.00}$ | $\mathbf{92.53 \pm 0.00}$ | $\mathbf{24.37 \pm 0.00}$ | $\mathbf{41.59 \pm 0.00}$ | $17.22 \pm 0.00$ |
| | GLM-4v-9B (Wang et al., 2023) | $38.13 \pm 0.06$ | $19.96 \pm 0.05$ | $91.81 \pm 0.08$ | $18.50 \pm 0.35$ | $25.37 \pm 0.51$ | $6.87 \pm 0.84$ |
| | InternVL2-26B (Chen et al., 2024a) | $129.39 \pm 7.84$ | $16.75 \pm 0.15$ | $85.83 \pm 0.18$ | $14.13 \pm 1.13$ | $26.25 \pm 3.58$ | $12.12 \pm 4.70$ |
| | InternVL3-38B (Chen et al., 2024a) | $63.18 \pm 2.45$ | $20.29 \pm 0.26$ | $90.32 \pm 0.18$ | $17.09 \pm 1.07$ | $28.32 \pm 0.89$ | $11.23 \pm 1.47$ |
| | LLaVA15-13B (Liu et al., 2023) | $60.26 \pm 0.00$ | $10.74 \pm 0.00$ | $84.21 \pm 0.00$ | $7.75 \pm 0.00$ | $16.81 \pm 0.00$ | $9.06 \pm 0.00$ |
| | LLaVA-v16-13B (Liu et al., 2023) | $194.07 \pm 0.00$ | $12.13 \pm 0.00$ | $80.38 \pm 0.00$ | $12.82 \pm 0.00$ | $15.04 \pm 0.00$ | $3.99 \pm 0.00$ |
| | MiniCPM-V2-6B (Yao et al., 2024) | $106.41 \pm 4.72$ | $15.08 \pm 0.21$ | $85.52 \pm 0.34$ | $14.46 \pm 0.33$ | $25.07 \pm 0.51$ | $10.61 \pm 0.45$ |
| | Phi-4-MM-instruct (Abouelenin et al., 2025) | $52.78 \pm 0.00$ | $12.74 \pm 0.00$ | $87.72 \pm 0.00$ | $12.39 \pm 0.00$ | $19.47 \pm 0.00$ | $7.08 \pm 0.00$ |
| | Qwen25VL-32B (Bai et al., 2025) | $41.53 \pm 0.06$ | $20.37 \pm 0.09$ | $\mathbf{90.95 \pm 0.06}$ | $16.85 \pm 0.05$ | $34.22 \pm 0.51$ | $17.36 \pm 0.65$ |

Table 3: **Mean Absolute Error (MAE) over (a) different regions and (b) different periods.**

| Method | Model | (a) Regions (Continents, MAE, $\downarrow$) | | | | | (b) Period (MAE, $\downarrow$) | | | | | | |
|---|---|---|---|---|---|---|---|---|---|---|---|---|---|
| | | Africa | Americas | Asia | Australia | Europe | 1000–1150 | 1150–1400 | 1400–1600 | 1600–1800 | 1800–1900 | 1900–1950 | 1950–2024 |
| CNN | ResNet-50 (He et al., 2016) | 102.13 | 34.97 | 71.48 | 36.51 | 92.37 | 634.98 | 423.93 | 233.93 | 88.45 | 35.89 | 34.33 | 37.42 |
| | ConvNeXt-B (Liu et al., 2022) | 85.07 | 29.13 | 56.02 | 29.04 | 81.02 | 538.27 | 348.31 | 200.69 | 82.37 | 31.00 | 29.36 | 29.59 |
| Transformer | ViT-B/16 (Dosovitskiy et al., 2020) | 100.41 | 31.23 | 72.55 | 32.49 | 88.99 | 236.50 | 199.25 | 218.36 | 93.35 | 32.75 | 30.76 | 30.68 |
| | Swin-B (Liu et al., 2021) | 104.53 | 31.08 | 69.26 | 29.93 | 86.10 | 557.44 | 402.18 | 218.38 | 79.21 | 32.53 | 32.69 | 30.07 |
| CLIP-based | CLIP (zero-shot) (Li et al., 2021) | 148.98 | 53.72 | 116.45 | 64.07 | 125.09 | 228.83 | 215.68 | 185.49 | 99.41 | 71.56 | 81.51 | 54.12 |
| | GeoCLIP (Vivanco Cepeda et al., 2023) | 100.53 | 27.12 | 62.40 | 25.33 | 87.22 | 197.04 | 145.22 | 171.61 | 81.67 | 41.83 | 32.69 | 30.07 |
| | NumCLIP (Du et al., 2024) | 85.42 | 24.72 | 54.97 | 25.97 | 75.40 | 495.76 | 293.58 | 191.47 | 78.29 | 27.95 | 23.09 | 27.87 |
| | YearCLIP (ours) | 85.85 | 26.10 | 53.20 | 24.90 | 71.31 | 483.31 | 282.46 | 185.55 | 78.75 | 27.69 | 22.84 | 27.45 |
| Closed VLMs | GPT4o-mini (Achiam et al., 2023) | 98.50 | 30.50 | 51.73 | 28.62 | 71.83 | 402.97 | 238.45 | 198.05 | 101.32 | 35.92 | 22.52 | 20.92 |
| | Gemini1.5-pro (Gemini Team et al., 2024) | 67.71 | 27.91 | 60.17 | 30.22 | 66.27 | 386.86 | 226.83 | 157.73 | 70.83 | 26.22 | 16.88 | 21.40 |
| | Gemini2.0-flash (Gemini Team et al., 2025) | 62.73 | 23.53 | 39.31 | 20.91 | 57.80 | 273.82 | 175.35 | 142.04 | 71.39 | 30.22 | 18.45 | 20.43 |
| | Claude3-haiku (Anthropic Team, 2024) | 91.85 | 25.34 | 63.81 | 30.23 | 78.77 | 284.09 | 290.85 | 202.74 | 107.55 | 39.21 | 26.88 | 26.64 |
| | Grok2 (xAI Team, 2024) | 72.00 | 23.92 | 45.87 | 19.75 | 62.58 | 165.47 | 62.58 | 165.47 | 79.22 | 35.61 | 19.04 | 23.51 |
| OpenVLM | CogVLM2-19B (Hong et al., 2024) | 105.48 | 28.77 | 56.77 | 26.85 | 72.89 | 388.90 | 243.09 | 199.62 | 85.26 | 34.26 | 27.44 | 23.35 |
| | Gemma3-27B (Team, 2025) | 89.52 | 24.38 | 48.36 | 21.07 | 63.35 | 349.20 | 231.03 | 166.36 | 75.11 | 29.89 | 19.78 | 22.68 |
| | GLM-4v-9B (Wang et al., 2023) | 82.00 | 27.16 | 49.96 | 23.47 | 65.21 | 190.07 | 163.48 | 141.96 | 75.48 | 30.92 | 24.55 | 22.63 |
| | InternVL2-26B (Chen et al., 2024a) | 163.84 | 52.50 | 114.88 | 275.11 | 184.64 | 539.88 | 418.39 | 346.32 | 48.03 | 121.22 | 93.48 | 102.90 |
| | InternVL3-38B (Chen et al., 2024a) | 69.69 | 26.15 | 77.35 | 33.80 | 94.90 | 380.35 | 251.13 | 188.98 | 110.02 | 53.76 | 47.46 | 51.63 |
| | LLaVA15-13B (Liu et al., 2023) | 88.52 | 35.67 | 50.00 | 40.00 | 95.00 | 508.39 | 374.03 | 143.05 | 88.25 | 16.69 | 26.26 | 22.85 |
| | LLaVA-v16-13B (Liu et al., 2023) | 368.85 | 42.07 | 232.46 | 255.60 | 95.00 | 558.97 | 370.10 | 145.15 | 100.20 | 16.96 | 177.63 | 121.94 |
| | MiniCPM-V2-6B (Yao et al., 2024) | 205.98 | 33.83 | 88.98 | 116.66 | 147.21 | 456.59 | 271.43 | 144.95 | 84.76 | 30.86 | 76.92 | 144.65 |
| | Phi-4-MM-instruct (Abouelenin et al., 2025) | 81.68 | 30.16 | 60.61 | 35.56 | 73.40 | 499.13 | 231.37 | 199.49 | 57.16 | 37.86 | 32.84 | 33.68 |
| | Qwen25VL-32B (Bai et al., 2025) | 61.08 | 30.64 | 42.64 | 26.46 | 68.44 | 411.36 | 199.96 | 192.55 | 94.20 | 36.95 | 21.22 | 21.76 |

## 5.2 POPULARITY ANALYSIS

Table 2 shows $IA_5$ across five popularity bins based on Wikipedia page views. *Gain* is the difference in $IA_5$ between high ($>10^5$) and low popularity ($<10^2$) buildings.

**General trend across models.** Across CNNs, Transformers, and CLIP-based methods, highly popular buildings often yield worse accuracy. For example, ConvNeXt-B (CNN) drops from 16.57% to 12.68% (Gain: -3.89%), Swin-B (Transformer) from 15.82% to 6.78% (-9.04%), and YearCLIP (CLIP-based) from 20.19% to 12.39% (-7.80%). This consistent decline suggests that iconic landmarks are harder to date, likely due to stylistic heterogeneity, renovations, or multiple historical narratives.

**Popularity bias in VLMs.** Both closed and open source VLMs display the opposite pattern, with consistently higher scores on popular buildings. Gemini2.0-Flash jumps from 24.23% to 58.41% (+34.18%), Gemini1.5-Pro from 26.76% to 43.36% (+16.60%), Grok2 from 25.77% to 42.48% (+16.71%), and Qwen2.5VL-32B from 16.85% to 34.22% (+17.36%). However, such gain reflects a strong popularity bias. Models likely exploit memorized associations from pre-training data, rather than demonstrating genuine architectural reasoning. This undermines their reliability, as performance becomes inflated on well-documented landmarks while offering little insight into less-known or underrepresented buildings.

Table 4: **Mean Absolute Error (MAE) over different population density and renovation types.**

| Method | Model | Population Density (MAE, ↓) | | | Renovation (MAE, ↓) | | |
|---|---|---|---|---|---|---|---|
| | | < 300 | 300–1500 | > 1500 | Never | Renovated | Rebuilt |
| CNN | ConvNeXt-B (Liu et al., 2022) | 47.15 | 40.63 | 44.32 | 33.11 | 46.50 | 68.46 |
| Transformer | Swin-B (Liu et al., 2021) | 51.17 | 43.59 | 47.24 | 37.09 | 50.30 | 70.82 |
| CLIP-based | YearCLIP (ours) | 42.67 | 36.22 | 39.04 | 30.62 | 41.84 | 59.04 |
| Closed VLMs | Gemini1.5-pro (Gemini Team et al., 2024) | **37.11** | **30.07** | **32.07** | **20.66** | **34.99** | **57.36** |
| OpenVLM | Gemma3-27B (Team, 2025) | 40.49 | 32.63 | 35.76 | 24.64 | 37.96 | 61.13 |

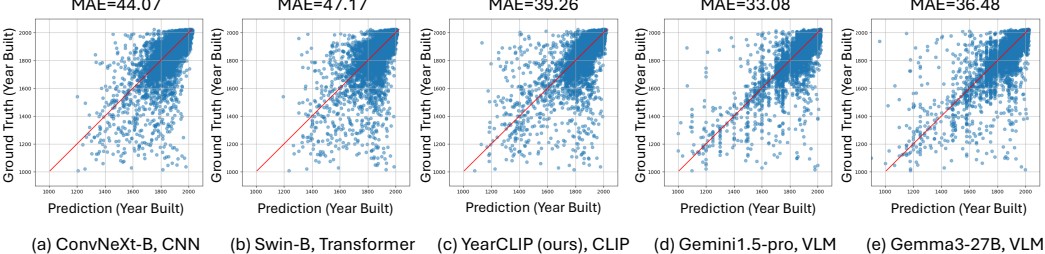

(a) ConvNeXt-B, CNN  (b) Swin-B, Transformer  (c) YearCLIP (ours), CLIP  (d) Gemini1.5-pro, VLM  (e) Gemma3-27B, VLM

Figure 5: **Prediction error scatter plots for representative models.** (a) ConvNeXt-B (CNN), (b) Swin-B (Transformer), (c) YearCLIP (ours, CLIP-based), (d) Gemini1.5-pro (VLM), and (e) Gemma3-27B (VLM). The horizontal axis shows predicted construction year, vertical axis shows groundtruth. Each point represents a single building. The red diagonal line indicates perfect prediction.

## 5.3 REGIONAL AND PERIOD ANALYSIS

Table 3 reports MAE across five continents and eight historical periods, highlighting geographic and temporal biases.

**Geographic Biases.** Nearly every methods exhibit clear regional disparities: Americas and Australia yield the lowest MAE, while Africa and Europe are highest, with Asia in between. For example, Gemini2.0-flash achieves MAE = 23.53 (Americas) vs. 62.73 (Africa) and 57.80 (Europe). Similar trends appear in NumCLIP (Asia: 58.97, Europe: 71.31), reflecting skewed pre-training data.

**Americas Dominance and YearCLIP.** GeoCLIP, NumCLIP, and YearCLIP (ours) reach their lowest MAE in Americas (27.12, 26.97, 26.10), aligned with the dataset's Americas-heavy distribution (63.3%, Fig. 3). YearCLIP reduces this bias, achieving more balanced results across regions.

**Temporal Trends and Early-Period Challenges.** Table 3(b) shows a clear temporal effect: models achieve markedly lower MAE in later periods (e.g., Gemini1.5-pro: 386.86 in 1000-1150 vs. 16.88 in 1900-1950). Performance degrades sharply for the earliest periods, where MAE often exceeds 300 across methods (e.g., Qwen25VL-32B: 411.36 vs. 21.22). This gap likely stems from data scarcity, preservation bias, and the greater stylistic heterogeneity of ancient architecture, whereas abundant examples of modern buildings provide richer training signals.

## 5.4 IMPACT OF POPULATION DENSITY AND RENOVATION STATUS

Table 4 shows MAE by population density and renovation status. Population density, collected via GPWv4.11 API, is categorized as **Rural** (< 300 people/km$^2$, 27.33%), **Semi-urban** (300–1500, 26.30%), and **Urban** (> 1500, 46.37%). Renovation status, derived from 11,087 Wikipedia descriptions via LLM, includes **Never** (13.65%), **Renovated** (52.99%), and **Rebuilt** (0.25%).

**Population Density.** Semi-urban regions yield the lowest MAE across most models (YearCLIP: 36.22, Gemini1.5-pro: 30.07), while rural and urban areas are harder, likely due to architectural variety or mixed-era skylines. For example, Gemma3-27B: Rural 40.49 vs. Urban 35.76.

**Renovation Status.** MAE is lowest for never-renovated buildings (Gemini1.5-pro: 20.66), increases for renovated ones (34.99), and peaks for rebuilt cases (57.36). Reconstructions may erase original architectural cues, making year prediction unreliable.

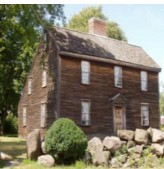

The estimated built year is **1880 (GT=1868)**, as the architectural style seems to be **Neoclassical and Industrial Age**. The level of ornamentation is **highly ornate**. The roof of the building is a **spire**. The window of the building is a **curtain window**. The structural shape of the building is **irregular layout**. ~~The wall of the building is made of timber wall~~.

The estimated built year is **1708 (GT=1681)**, as the architectural style seems to be **Baroque and Rococo Architecture**. The wall of the building is a **simple, flat wall**. The building exhibits a **symmetrical, and plain form**. ~~The material used for the roof is metal roofing~~. The building has ~~decorative columns~~ used for aesthetic effect. The wall of the building is made of **timber and wooden elements**.

Figure 6: **Explainable age predictions with YEARCLIP.** Powered by Reason-enhanced NumCLIP, the system predicts **construction year** within ±15 yr of **ground truth** and provides rationales that highlight **stylistic** and **historic** cues. CLIP baselines miss or misassign these signals, whereas our location + reason pipeline yields transparent, verifiable explanations.

## 5.5 PREDICTION DISTRIBUTION

Figure 5 compares the prediction errors of five models, including ConvNeXt-B, Swin-B, YearCLIP, Gemini1.5-pro, and Gemma3-27B. CNN (ConvNeXt-B) and Transformer (Swin-B) show larger deviations, especially for pre-1600 buildings. YearCLIP yields predictions closer to the diagonal, indicating higher accuracy across periods. VLMs (Gemini1.5-pro, Gemma3-27B) cluster tightly along the diagonal, benefiting from linguistic and geographic context, though accuracy remains higher for post-1800 buildings due to temporal imbalance in training data.

## 5.6 EXPLAINABILITY

Figure 6 shows the output of YearCLIP, a Reason-enhanced NumCLIP with additional location conditions. The explainable age prediction can reduce the MAE and provide a human-verifiable rationale for each prediction. The reasoning system identifies architectural features such as roof types, materials, and structural elements that most strongly influence the predicted construction year, offering transparency into the model's decision-making process in comparison to NumCLIP.

## 6 CONCLUSION

We presented YEARGUESSR, the first CC BY-SA 4.0, large-scale dataset and benchmark for building-age estimation, plus an ordinal protocol, a popularity-aware metric, and a 30-model study that reveals landmark-memorization bias while showing our NumCLIP-Loc system halves MAE over decade classification. The resource can aid heritage preservation, retrofit planning, and disaster inspection, yet it might also help locate vulnerable sites or reinforce regional bias. Safeguards include removal of non-free images, an accompanying responsible-use data card, and public bias metrics; closed-source VLMs are accessed via documented prompts without redistributing weights.

**Limitations.** The dataset is geographically (63% Americas, 22.5% Western Europe) and temporally skewed toward modern examples, with uneven rural–urban coverage, limiting generalization to underrepresented regions and early styles. Labels are based on original construction years from Wikipedia/Wikimedia, even for buildings that have been substantially renovated or rebuilt, introducing noise that hampers accurate visual-year estimation.

**Future Work.** We aim to address these issues by expanding non-Western and early-period coverage (e.g., integrating CMAB (Zhang et al., 2024b) and other regional datasets, targeted collection for low-resource regions), digitizing pre-1600 records from museums and archives, and adding explicit renovation/temporal-segmentation labels. To mitigate data scarcity, we plan to fine-tune diffusion priors for synthetic augmentation (Trabucco et al., 2023), and explore active learning, debiasing, and expert-in-the-loop validation to improve robustness, fairness, and reproducibility.

## 7 ETHICS STATEMENT

**Dataset Bias:** Our dataset exhibits geographical bias toward Western architecture (85.8% from Americas and Europe, Figure 3(a)), potentially underrepresenting non-Western architectural traditions. We provide continent-specific metrics (Table 3) to highlight performance disparities and recommend expanding coverage of underrepresented regions.

**Potential Misuse:** While supporting heritage preservation and sustainability, this technology could be misused for property speculation, insurance discrimination, or targeting vulnerable structures. We recommend implementing safeguards for responsible deployment.

**VLMs Bias:** VLMs show popularity bias with up to +34.18% accuracy gain on famous landmarks (Section 5.2, Table 2), suggesting memorization rather than architectural understanding. This could lead to inequitable outcomes favoring well-documented buildings over culturally significant but less-publicized structures.

**Data Source:** All images are from publicly available Wikipedia/Wikimedia Commons under CC BY-SA 4.0 licensing (Section 3.1). Users should consider privacy implications when applying similar techniques to private datasets.

**Cultural Sensitivity:** Automated age estimation may oversimplify complex architectural histories, particularly for renovated buildings (Figure 3(d)). Users should consult domain experts when dealing with culturally significant heritage sites.

## 8 REPRODUCIBILITY STATEMENT

To ensure reproducibility of our work, we provide the following resources and implementation details:

**Dataset:** The YearGuessr dataset comprising 55,546 Wikipedia images with construction years, GPS coordinates, and other metadata will be made publicly available under CC BY-SA 4.0 licensing. Complete dataset construction details, including the data collection pipeline, cleaning procedures, and train/validation/test splits, are described in Section 3. Dataset statistics and distributions are provided in Section 3 and Figure 3.

**Model Implementation:** Complete source code for the YearCLIP model, including the architecture described in Figure 4 and Section 4, is provided in Appendix. Training hyperparameters, loss functions, and optimization details are specified in Appendix A. The complete training pipeline with coarse-to-fine ordinal regression is detailed in Appendix B, including input processing (Appendix B.2), pre-defined elements (Appendix B.3), and loss function implementation (Appendix B.7).

**Experimental Setup:** All experimental configurations, including hardware specifications (NVIDIA RTX 4090 GPU), training parameters, and evaluation protocols are documented in Appendix A. Evaluation metrics and protocols are defined in Section 3.3. We report results with standard deviations across three random seeds for all experiments.

**Reasoning Prompts:** All predefined reasoning prompts used for architectural feature categorization, including roof types, materials, and structural elements, are detailed in Table 5 and Appendix A. The reasoning importance analysis methodology is described in Appendix B.6.

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

APPENDIX

## A    IMPLEMENTATION DETAILS

**Experimental setup.** We split the dataset into training, validation, and test sets with 33,337, 11,122, and 11,087 samples, respectively, following a 6:2:2 ratio. All experiments are conducted on an NVIDIA RTX 4090 GPU. We use the following hyperparameters: learning rates for the image encoder and adapter are set to $1 \times 10^{-5}$, while the optimizer uses RAdam with a learning rate of $1 \times 10^{-4}$, and Adam betas of 0.9 and 0.999. The learning rate scheduler is multi-step with a step size of 60 and a gamma of 0.1. Loss weights are balanced with cross-entropy, KL divergence, and regression terms, each set to 1.0. We set the number of ranks to 7, use ViT-B/16 for both text and image encoders, and train with a batch size of 64 for 50 epochs at 16-bit precision.

**Predefined Reasoning.** In the experimental design, predefined prompts were generated with the assistance of the Gemini2.0 (Gemini Team et al., 2025) model to facilitate the analysis of building age estimation. These prompts were crafted to produce textual rationales potentially relevant to architectural age judgment, subsequently categorized into distinct options. For instance, considering the building's roof as a judgment criterion, Gemini2.0 (Gemini Team et al., 2025) segmented the roof types into specific categories, including spire, dome, flat roof, sloped roof, gabled roof, mansard roof, and butterfly roof. An illustrative example is provided below table 5:

Table 5: **Example roof types and their descriptions for building age estimation.**

| Roof Type | Description |
| --- | --- |
| spire | A sharply pointed roof emphasizing verticality and ornate detailing. |
| dome | A smoothly curved roof suggesting grandeur and centrality. |
| flat roof | A completely horizontal surface with an unobstructed and minimalist design. |
| sloped roof | A roof with a noticeable and functional inclination for water drainage and dynamic appearance. |
| gabled roof | A traditional peaked roof with a triangular profile that exudes symmetry. |
| mansard roof | A dual-pitched roof offering both elegance and additional living space. |
| butterfly roof | An inverted roof design that creates a V-shaped, modern, and unconventional look. |

Beyond the roof, additional judgment criteria can be incorporated based on the task requirements, enabling further customization. Such enhancements enrich the input to the regressor, potentially improving the model's performance and adaptability for architectural age estimation tasks.

## B    TRAINING PIPELINE

This section outlines the training pipeline for the YearCLIP architecture, which predicts the construction year of buildings using a coarse-to-fine approach. The pipeline is divided into five parts, preceded by an introduction to define the inputs and pre-defined elements.

### B.1    INTRODUCTION

The YearCLIP model in Figure 7 takes an image of a building and optional geographic coordinates as inputs. The image is a 224×224 pixel color image, while the coordinates are a pair of latitude and longitude values, which might not always be available. We also use pre-defined elements, including seven architectural styles (like Roman or Gothic), each linked to a specific historical period, and a set of reason prompts (like roof type or building material), where each prompt has subcategories (e.g., roof types include spire, dome, flat roof, etc.). The goal is to predict the building's construction year and explain the reasoning behind the prediction. The model uses several components: an image encoder, a location encoder, a class encoder for styles, a reason encoder for prompts, and a regressor to make the final prediction.

Figure 7: **YEARCLIP architecture.** An image encoder $f_v$ (CLIP) extracts $224\times224$ facade features. We then fuse the feature with a GPS embedding from the location encoder $f_l$ (RFF + MLP, optional input) via a learnable zero-convolution. Parallel text branches encode (i) seven coarse style classes $f_c$ and (ii) a bank of reasoning prompts $f_r$ describing roofs, walls, heights, etc. All frozen encoders feed a trainable regressor $g(\cdot)$ that performs coarse-to-fine ordinal regression. It outputs the predicted construction year (here 1687), selects the best-fit style/reason tokens, and outputs a readable rationale.

## B.2  INPUT PROCESSING

We process the input data in the following steps:

1. **Image Encoding:** The image encoder, based on CLIP, processes the building image to extract raw visual features:

$$\mathbf{z}_v^{\text{raw}} = f_v(I), \tag{1}$$

where $I$ is the input image, and $f_v$ is the image encoder. These raw features are then passed through a multi-layer perceptron (MLP) to obtain the final image embedding:

$$\mathbf{z}_v = \text{MLP}(\mathbf{z}_v^{\text{raw}}). \tag{2}$$

2. **Location Encoding:** If geographic coordinates are provided, the location encoder first transforms them into raw features via an MLP:

$$\mathbf{z}_l^{\text{raw}} = f_l(L), \tag{3}$$

where $L$ is the pair of latitude and longitude values. These features are then processed by a learnable zero-convolution to produce the final location embedding:

$$\mathbf{z}_l = \text{ZeroConv}(\mathbf{z}_l^{\text{raw}}), \tag{4}$$

where ZeroConv ensures proper alignment for subsequent fusion.

3. **Combining Embeddings:** When coordinates are available, we combine the image embedding and location embedding by directly adding them:

$$\mathbf{z}_{\text{input}} = \mathbf{z}_v + \mathbf{z}_l, \tag{5}$$

where the addition is performed element-wise. If coordinates are missing, the input embedding is simply the image embedding $\mathbf{z}_v$. This allows the model to work even without location data.

### B.3 PRE-DEFINED ELEMENTS

We define and process the pre-defined elements as follows:

1. **Building Styles Encoding:** We have seven architectural styles, each tied to a historical period: Roman (800–1150), Gothic (1150–1400), Renaissance (1400–1600), Baroque (1600–1750), Neoclassical (1750–1850), Modern (1850–1950), and Contemporary (1950–present). The class encoder processes each style to produce an embedding:

$$\mathbf{z}_{c_i} = f_c(s_i), \quad i = 1, 2, \ldots, 7, \tag{6}$$

   where $s_i$ is a style (e.g., Roman), and $f_c$ is the class encoder. This results in a set of embeddings for all seven styles.

2. **Reason Prompts Encoding:** We use reason prompts like roof type, material, and height, each with subcategories (e.g., roof types include spire, dome, flat roof, etc.; materials include stone, brick, etc.). The reason encoder processes each subcategory to create an embedding:

$$\mathbf{z}_{r_{jk}} = f_r(r_{jk}), \tag{7}$$

   where $r_{jk}$ is a subcategory (e.g., spire for roof type), and $f_r$ is the reason encoder. This forms a collection of embeddings for all subcategories.

### B.4 COARSE STAGE

We measure how well the input matches the pre-defined elements by computing similarities:

1. **Style Similarity:** We calculate the similarity between the input embedding and each style embedding using cosine similarity:

$$\mathrm{sim}_{c_i} = \mathrm{sim}(\mathbf{z}_{\text{input}}, \mathbf{z}_{c_i}), \quad i = 1, 2, \ldots, 7, \tag{8}$$

   where sim measures the cosine similarity between two embeddings, resulting in a set of scores showing how well the input matches each style.

2. **Reason Similarity:** Similarly, we compute the similarity between the input embedding and each reason subcategory embedding:

$$\mathrm{sim}_{r_{jk}} = \mathrm{sim}(\mathbf{z}_{\text{input}}, \mathbf{z}_{r_{jk}}), \tag{9}$$

   forming a set of scores for all subcategories.

### B.5 FINE REGRESSION

The regression process uses the similarity scores to predict the construction year:

1. **Preparing Input for Regressor:** We combine the style and reason similarity scores into a single vector for the regressor:

$$\mathbf{s} = [\mathrm{sim}_{c_1}, \mathrm{sim}_{c_2}, \ldots, \mathrm{sim}_{c_7}, \mathrm{sim}_{r_{11}}, \mathrm{sim}_{r_{12}}, \ldots, \mathrm{sim}_{r_{m_n}}], \tag{10}$$

   where the vector includes all style similarities and all reason subcategory similarities.

2. **Computing Probabilities:** The regressor processes this vector to produce probabilities for each of the seven historical periods, indicating the likelihood that the building belongs to each period.

3. **Final Prediction:** The final predicted year is calculated as a weighted average of the midpoints of the historical periods, adjusted by the probabilities and a small confidence term:

$$\hat{y} = \sum_{i=1}^{k} p_i \cdot \frac{b_i}{1 + \delta_i}, \tag{11}$$

   where $p_i$ is the probability for the $i$-th period, $b_i$ is the midpoint of that period, and $\delta_i$ is a small learnable parameter for stability.

### B.6 REASONING IMPORTANCE

We analyze the importance of each reason to explain the prediction:

1. **Subcategory Importance:** We calculate an importance score for each reason subcategory by combining its similarity score with the regressor's attention to the corresponding historical period.

2. **Selecting Key Subcategories:** For each reason (like roof or material), we pick the subcategory with the highest importance score. We also sum the importance scores of all subcategories for each reason to find the overall importance of that reason.

3. **Top Reasons:** We select the top five reasons with the highest overall importance, providing insights into the key factors (e.g., roof type: dome, material: stone) that influenced the predicted year.

### B.7 LOSS FUNCTION

To train the model, we use a combination of two loss terms:

1. **Fine-grained Cross-modal Ranking-based Contrastive Loss (FCRC):** Following Num-CLIP (Du et al., 2024), we adopt a ranking-based contrastive loss to enforce ordinal consistency between predicted and ground-truth labels. The FCRC loss is defined as:

$$\mathcal{L}_{\text{FCRC}}^{z} = -\sum_{i=1}^{M} \frac{1}{M} \log \left[ \frac{f(z^i, w^i)}{f(z^i, w^i) + \sum_{j \neq i} \lambda_{i,j}^{z} \, f(z^i, w^j)} \right], \quad (12)$$

where $f(z^i, w^j) = \exp(\cos(z_i, w_j)/\tau)$ measures the similarity between image embedding $z_i$ and text embedding $w_j$, and $\lambda_{i,j}^{z}$ denotes the regularisation weight of the $j$-th negative sample.

2. **Weighting of Negative Samples:** The weight parameter $\lambda_{i,j}$ is determined by the label distance between samples:

$$\lambda_{i,j} = \text{Norm}(\beta \cdot d_{i,j}), \quad d_{i,j} = |y_i - y_j|, \quad (13)$$

where $y_i$ and $y_j$ are the ground-truth labels of the anchor and negative samples, $\beta$ is a scaling factor, and Norm ensures the weights sum to 1.

## C FULL TABLE RESULTS

### C.1 BASIC METHOD ON BUILDING YEAR ESTIMATION

To evaluate the performance of different architectural models for building age estimation on the YearGuessr dataset (Dionelis et al., 2025), we conducted a comparative analysis based on the Mean Absolute Error (MAE) metric, as presented in Figure 8. This figure illustrates the MAE values achieved by a diverse set of models, categorized into three groups: CNN-based (green) (He et al., 2016; Liu et al., 2022), Transformer-based (blue) (Dosovitskiy et al., 2020; Liu et al., 2021), and CLIP-based (yellow) methods. The MAE values, ranging from 40 to 60, are plotted along a horizontal axis, with each model labeled at its corresponding MAE position. Notable CNN-based models include ResNet50 (MAE 54.59) (He et al., 2016), ResNet152 (MAE 47.70) (He et al., 2016), and ConvNeXt-L (MAE 42.34) (Liu et al., 2022), demonstrating a trend of improved accuracy with increased model complexity. Transformer-based models, such as ViT-B/16 (MAE 48.86) (Dosovitskiy et al., 2020) and Swin-B (MAE 47.71) (Liu et al., 2021), show competitive performance, while CLIP-based models like GeoCLIP (MAE 44.32) (Vivanco Cepeda et al., 2023) and NumCLIP (MAE 40.01) (Du et al., 2024) exhibit the lowest MAE values, highlighting the efficacy of vision-language integration. This comparison underscores the trade-offs between model complexity, architectural design, and prediction accuracy, providing insights for selecting appropriate models for future tasks.

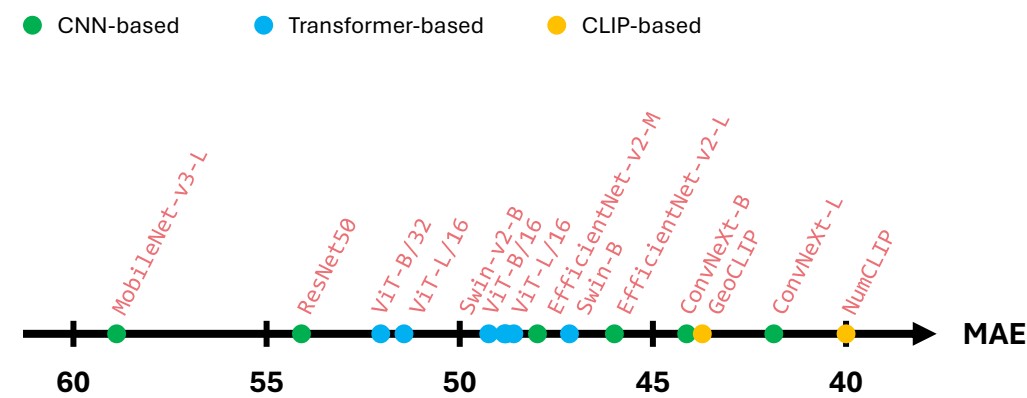

Figure 8: **Comparison of basic methods based on Mean Absolute Error (MAE).** The figure displays MAE values for various models, categorized by method type: CNN-based (green), Transformer-based (blue), and CLIP-based (yellow). Methods are positioned along the MAE axis, ranging from 40 to 60, with labels indicating model names.

## C.2 FULL RESULTS OF CLIP-BASED MODEL AND VLMS

The following tables (6, 7, and 8) provide the complete experimental results corresponding to Results and Anaysis in the main paper. These tables extend the analysis by including a broader range of models, encompassing various model sizes and architectures. Specifically, Table 6 reports performance on basic metrics (Mean Absolute Error and Classification Accuracy) and interval accuracy across different year ranges. Table 7 details interval accuracy within ±5 years across popularity intervals, while Table 8 presents the same metric across different continents. These comprehensive results offer deeper insights into the performance of models with varying capacities on the building age estimation task.

## D LLM USAGE STATEMENT

This work involved the use of Large Language Models (LLMs) in several limited capacities, none of which reached the level of contribution that would warrant co-authorship. The specific uses are detailed below:

**Predefined Reasoning Prompt Generation (Section 4, Appendix A):** Gemini2.0 (Gemini Team et al., 2025) was used to generate predefined reasoning prompts for architectural age estimation analysis. Specifically, the model helped categorize building features into structured prompts, such as roof types (spire, dome, flat roof, sloped roof, gabled roof, mansard roof, butterfly roof) and their corresponding descriptions as shown in Table 5. The LLM assisted in creating comprehensive categorical descriptions for various architectural elements including materials, heights, and structural features that serve as input to our reasoning system.

**Dataset Annotation Assistance (Section 3):** An LLM was employed to analyze building descriptions from Wikipedia to extract renovation status information, as mentioned in the renovation scenario analysis (Figure 3(d)). The LLM helped distinguish between *renovated* buildings (where original construction year remains valid) and *rebuilt* buildings (where construction year is redefined) from textual descriptions in the dataset.

**Writing and Code Assistance:** LLMs were used as general-purpose writing assistance tools for improving clarity, grammar, and style throughout the manuscript. Additionally, LLMs provided coding assistance for data processing, experimentation scripts, and visualization code. However, all core algorithmic contributions, experimental design, analysis, and scientific insights were conceived and developed by the human authors.

**Scope of Usage:** The LLM usage was limited to auxiliary tasks and did not involve research ideation, hypothesis generation, experimental design, result interpretation, or the core technical contributions of the paper. All scientific claims, methodological innovations, and experimental conclusions remain

Table 6: **Performance on Basic metrics and Accuracy within intervals.**

| Method | Basic | | Interval Accuracy (IA) | | | |
|---|---|---|---|---|---|---|
| | MAE (↓) | CLS Acc (↑) | ≤ 5 yrs (↑) | ≤ 20 yrs (↑) | ≤ 50 yrs (↑) | ≤ 100 yrs (↑) |
| CLIP (zero-shot) (Radford et al., 2021) | 78.23 | 55.43 | 12.78 | 36.81 | 58.05 | 78.55 |
| GeoCLIP (Vivanco Cepeda et al., 2023) | 44.32 | **70.16** | **24.48** | **56.25** | 77.80 | 89.57 |
| NumCLIP (Du et al., 2024) | 40.01 | 69.12 | 19.35 | 54.40 | **79.53** | 91.76 |
| YearCLIP (ours) | **39.38** | 68.67 | 18.50 | 54.17 | 79.52 | **91.85** |
| GPT4o-mini (Achiam et al., 2023) | 42.69 | 68.95 | 22.75 | 54.51 | 76.43 | 89.62 |
| Gemini1.5-pro (Gemini Team et al., 2024) | **33.08** | **74.09** | 28.18 | **63.50** | **83.27** | **93.14** |
| Gemini2.0-flash (Gemini Team et al., 2025) | 33.91 | 73.50 | **29.71** | 62.61 | 82.46 | 92.75 |
| Claude3-haiku (Anthropic Team, 2024) | 47.88 | 62.03 | 16.13 | 47.42 | 73.21 | 88.47 |
| Grok2 (xAI Team, 2024) | 35.28 | 73.74 | 27.57 | 61.31 | 81.90 | 93.02 |
| CogVLM2-19B (Hong et al., 2024) | 41.50 | 66.74 | 18.39 | 52.34 | 77.65 | 90.74 |
| Gemma3-4B (Team, 2025) | 40.97 | 68.05 | 22.78 | 55.14 | 78.76 | 91.67 |
| Gemma3-12B (Team, 2025) | 36.97 | 70.67 | 24.82 | 58.10 | 80.71 | 92.28 |
| Gemma3-27B (Team, 2025) | **36.48** | 70.83 | **25.58** | **58.46** | **81.28** | **92.53** |
| GLM-4v-9B (Wang et al., 2023) | 38.27 | **70.85** | 20.03 | 55.60 | 79.79 | 91.73 |
| InternVL2-2B (Chen et al., 2024a) | 145.68 | 48.75 | 9.73 | 31.92 | 58.97 | 79.28 |
| InternVL2-4B (Chen et al., 2024a) | 67.15 | 55.46 | 12.00 | 37.76 | 66.91 | 85.88 |
| InternVL2-8B (Chen et al., 2024a) | 196.69 | 52.94 | 11.62 | 36.36 | 62.08 | 78.44 |
| InternVL2-26B (Chen et al., 2024a) | 77.22 | 63.49 | 16.84 | 47.79 | 73.38 | 87.71 |
| InternVL3-2B (Chen et al., 2024a) | 101.10 | 52.98 | 10.98 | 35.39 | 64.72 | 83.70 |
| InternVL3-8B (Chen et al., 2024a) | 54.24 | 60.24 | 16.20 | 46.26 | 73.39 | 89.60 |
| InternVL3-9B (Chen et al., 2024a) | 58.95 | 60.71 | 15.60 | 45.60 | 72.50 | 88.36 |
| InternVL3-14B (Chen et al., 2024a) | 63.35 | 59.61 | 16.61 | 46.32 | 72.10 | 88.22 |
| InternVL3-38B (Chen et al., 2024a) | 61.79 | 68.47 | 20.37 | 53.06 | 77.40 | 90.19 |
| LLaVA15-7B (Liu et al., 2023) | 51.05 | 59.48 | 13.46 | 44.40 | 72.09 | 87.50 |
| LLaVA15-13B (Liu et al., 2023) | 60.08 | 61.52 | 10.72 | 36.26 | 66.76 | 84.24 |
| LLaVA-v16-7B (Liu et al., 2023) | 52.01 | 60.32 | 13.68 | 42.48 | 71.39 | 88.14 |
| LLaVA-v16-13B (Liu et al., 2023) | 169.21 | 58.06 | 12.34 | 39.83 | 65.82 | 81.69 |
| MiniCPM-V2-6B (Yao et al., 2024) | 107.01 | 60.02 | 15.09 | 45.03 | 69.99 | 85.75 |
| Phi-4-MM-instruct (Abouelenin et al., 2025) | 52.78 | 55.32 | 12.74 | 42.10 | 70.86 | 87.72 |
| Qwen25VL-3B (Bai et al., 2025) | 91.38 | 63.57 | 16.54 | 49.42 | 74.46 | 88.63 |
| Qwen25VL-7B (Bai et al., 2025) | 40.06 | 70.21 | 19.88 | 53.86 | 77.42 | 91.10 |
| Qwen25VL-32B (Bai et al., 2025) | 41.66 | 67.02 | 20.31 | 52.97 | 77.05 | 90.91 |

solely the intellectual contribution of the human authors. The authors take full responsibility for all content, including any LLM-generated text that has been reviewed and validated.

Table 7: **Interval Accuracy within ±5 years over different popularity intervals.**

| Method | Views (Popularity) | | | | |
|---|---|---|---|---|---|
| | $< 10^2$ (↑) | $10^2$–$10^3$ (↑) | $10^3$–$10^4$ (↑) | $10^4$–$10^5$ (↑) | $> 10^5$ (↑) |
| CLIP (zero-shot) (Radford et al., 2021) | 13.52 | 12.38 | 13.86 | 11.49 | 7.96 |
| GeoCLIP (Vivanco Cepeda et al., 2023) | **25.49** | **23.60** | **22.92** | **19.67** | **20.35** |
| NumCLIP (Du et al., 2024) | 22.25 | 20.65 | 18.03 | 14.25 | 16.81 |
| YearCLIP (ours) | 20.42 | 19.65 | 17.11 | 16.02 | 9.73 |
| GPT4o-mini (Achiam et al., 2023) | 19.01 | 21.25 | 23.82 | 27.96 | 48.67 |
| Gemini1.5-pro (Gemini Team et al., 2024) | **26.76** | **27.26** | 28.54 | 32.04 | 43.36 |
| Gemini2.0-flash (Gemini Team et al., 2025) | 24.23 | 26.85 | **31.90** | **40.33** | **58.41** |
| Claude3-haiku (Anthropic Team, 2024) | 18.45 | 15.27 | 16.24 | 17.35 | 32.74 |
| Grok2 (xAI Team, 2024) | 25.77 | 25.98 | 29.55 | 29.50 | 42.48 |
| CogVLM2-19B (Hong et al., 2024) | 16.20 | 17.98 | 19.48 | 18.12 | 23.89 |
| Gemma3-4B (Team, 2025) | 20.56 | 22.39 | 23.76 | 22.87 | 25.66 |
| Gemma3-12B (Team, 2025) | **25.07** | 23.99 | 25.53 | 26.08 | 33.63 |
| Gemma3-27B (Team, 2025) | 24.37 | **24.12** | **26.86** | **28.95** | **41.59** |
| GLM-4v-9B (Wang et al., 2023) | 17.32 | 19.48 | 20.84 | 21.99 | 25.66 |
| InternVL2-2B (Chen et al., 2024a) | 9.01 | 10.01 | 9.35 | 9.83 | 9.73 |
| InternVL2-4B (Chen et al., 2024a) | 7.46 | 11.29 | 13.26 | 14.81 | 15.93 |
| InternVL2-8B (Chen et al., 2024a) | 8.87 | 11.17 | 12.13 | 13.37 | 20.35 |
| InternVL2-26B (Chen et al., 2024a) | 15.07 | 15.90 | 17.86 | 19.23 | 27.43 |
| InternVL3-2B (Chen et al., 2024a) | 7.89 | 10.16 | 12.33 | 12.60 | 18.58 |
| InternVL3-8B (Chen et al., 2024a) | 13.38 | 15.70 | 16.87 | 18.45 | 21.24 |
| InternVL3-9B (Chen et al., 2024a) | 15.07 | 15.22 | 15.98 | 15.80 | 25.66 |
| InternVL3-14B (Chen et al., 2024a) | 14.08 | 15.73 | 17.77 | 18.23 | 30.97 |
| InternVL3-38B (Chen et al., 2024a) | 17.46 | 19.36 | 21.59 | 23.76 | 24.78 |
| LLaVA15-7B (Liu et al., 2023) | 11.69 | 13.24 | 13.95 | 13.92 | 17.70 |
| LLaVA15-13B (Liu et al., 2023) | 7.61 | 9.64 | 12.71 | 11.71 | 18.58 |
| LLaVA-v16-7B (Liu et al., 2023) | 11.97 | 13.35 | 14.36 | 14.36 | 17.70 |
| LLaVA-v16-13B (Liu et al., 2023) | 13.10 | 11.99 | 12.91 | 11.93 | 15.04 |
| MiniCPM-V2-6B (Yao et al., 2024) | 14.37 | 14.91 | 15.28 | 15.14 | 24.78 |
| Phi-4-MM-instruct (Abouelenin et al., 2025) | 12.39 | 12.31 | 12.34 | 12.60 | 19.47 |
| Qwen25VL-3B (Bai et al., 2025) | 15.92 | 15.59 | 18.03 | 16.35 | 22.12 |
| Qwen25VL-7B (Bai et al., 2025) | 18.73 | 19.00 | 20.81 | 21.55 | 30.09 |
| Qwen25VL-32B (Bai et al., 2025) | 16.62 | 18.57 | 22.43 | 24.64 | 33.63 |

Table 8: **Interval Accuracy within ±5 years over different regions.**

| Method | Regions (Continents) | | | | |
|---|---|---|---|---|---|
| | Africa (↑) | Americas (↑) | Asia (↑) | Australia (↑) | Europe (↑) |
| CLIP (zero-shot) (Radford et al., 2021) | 12.30 | 13.29 | 20.60 | 12.46 | 10.44 |
| GeoCLIP (Vivanco Cepeda et al., 2023) | **15.57** | **26.99** | **23.08** | **25.84** | **13.94** |
| NumCLIP (Du et al., 2024) | 10.66 | 23.19 | 17.37 | 19.15 | 10.97 |
| YearCLIP (ours) | 13.11 | 22.04 | 20.10 | 16.41 | 10.40 |
| GPT4o-mini (Achiam et al., 2023) | 20.49 | 24.47 | 31.76 | 20.36 | 17.67 |
| Gemini1.5-pro (Gemini Team et al., 2024) | **25.41** | 30.54 | 30.77 | 31.61 | 22.04 |
| Gemini2.0-flash (Gemini Team et al., 2025) | **25.41** | **30.67** | **38.46** | **32.52** | **26.04** |
| Claude3-haiku (Anthropic Team, 2024) | 11.48 | 17.72 | 19.11 | 17.63 | 11.90 |
| Grok2 (xAI Team, 2024) | **25.41** | 29.85 | 31.76 | 31.31 | 21.21 |
| CogVLM2-19B (Hong et al., 2024) | 15.57 | 20.40 | 23.57 | 13.37 | 13.74 |
| Gemma3-4B (Team, 2025) | 17.21 | 24.75 | 25.06 | 27.36 | 17.34 |
| Gemma3-12B (Team, 2025) | 19.67 | 26.85 | 31.27 | 30.40 | 18.81 |
| Gemma3-27B (Team, 2025) | **26.23** | **27.41** | **32.75** | **31.91** | **19.61** |
| GLM-4v-9B (Wang et al., 2023) | 25.41 | 21.32 | 27.30 | 14.89 | 16.34 |
| InternVL2-2B (Chen et al., 2024a) | 7.38 | 10.44 | 13.15 | 7.90 | 7.87 |
| InternVL2-4B (Chen et al., 2024a) | 15.57 | 12.46 | 19.85 | 13.37 | 9.67 |
| InternVL2-8B (Chen et al., 2024a) | 14.75 | 12.18 | 14.89 | 16.72 | 8.97 |
| InternVL2-26B (Chen et al., 2024a) | 10.66 | 18.64 | 20.84 | 18.54 | 12.20 |
| InternVL3-2B (Chen et al., 2024a) | 10.66 | 11.19 | 19.85 | 13.07 | 8.94 |
| InternVL3-8B (Chen et al., 2024a) | 13.11 | 17.03 | 24.81 | 19.15 | 12.90 |
| InternVL3-9B (Chen et al., 2024a) | 11.48 | 16.50 | 22.08 | 17.93 | 12.40 |
| InternVL3-14B (Chen et al., 2024a) | 16.39 | 17.70 | 23.08 | 16.41 | 13.14 |
| InternVL3-38B (Chen et al., 2024a) | 13.11 | 21.86 | 30.02 | 25.23 | 15.21 |
| LLaVA15-7B (Liu et al., 2023) | 18.85 | 14.09 | 19.85 | 15.20 | 10.77 |
| LLaVA15-13B (Liu et al., 2023) | 18.03 | 10.31 | 21.59 | 10.94 | 9.94 |
| LLaVA-v16-7B (Liu et al., 2023) | 13.93 | 15.19 | 16.63 | 15.81 | 9.60 |
| LLaVA-v16-13B (Liu et al., 2023) | 13.11 | 13.85 | 14.14 | 14.89 | 8.24 |
| MiniCPM-V2-6B (Yao et al., 2024) | 13.93 | 16.34 | 19.60 | 17.02 | 11.37 |
| Phi-4-MM-instruct (Abouelenin et al., 2025) | 12.30 | 13.35 | 17.62 | 12.77 | 10.67 |
| Qwen25VL-3B (Bai et al., 2025) | 18.85 | 18.02 | 20.35 | 13.37 | 12.84 |
| Qwen25VL-7B (Bai et al., 2025) | 16.39 | 20.93 | 27.30 | 17.63 | 16.81 |
| Qwen25VL-32B (Bai et al., 2025) | 19.67 | 21.33 | 31.02 | 19.76 | 16.57 |

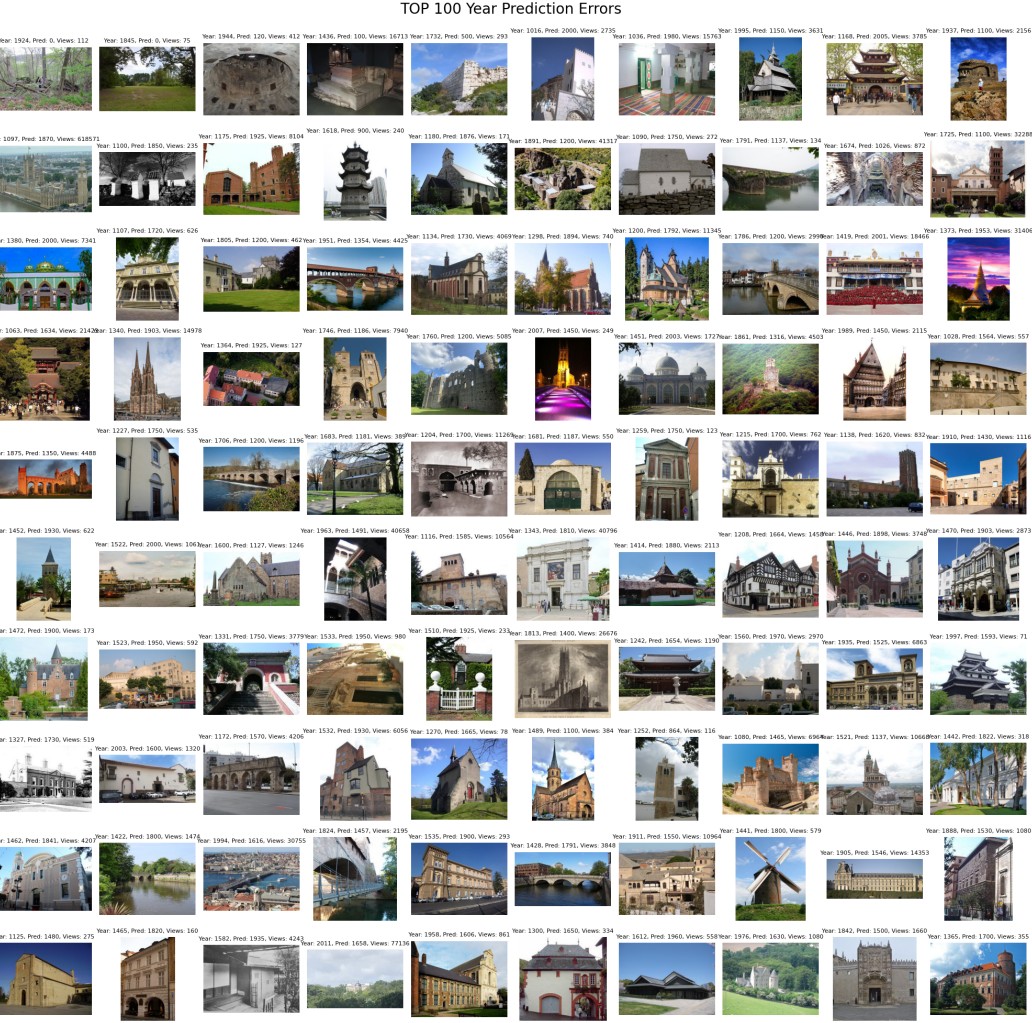

Figure 9: **Top 100 prediction errors by Gemini2.0-Flash.** This figure shows the 100 building images with the highest Mean Absolute Error (MAE) when predicted by Gemini2.0-Flash. Each image is labeled with the ground truth year (Year), predicted year (Pred), and Wikipedia page views (Views) as popularity indicator. These challenging cases illustrate common failure modes including ancient buildings, heavily renovated structures, and architecturally ambiguous facades.

