# OpenReview forum: "Exposing VLM Memorization of Famous Landmarks: A 55K Building Age Dataset Revealing Popularity Bias"
_ICLR.cc/2026/Conference — ICLR 2026 Conference Withdrawn Submission_

### Official Review · Reviewer_8873 · 2025-10-24

**Soundness:** 2
**Presentation:** 2
**Contribution:** 2
**Rating:** 4
**Confidence:** 3

**Summary:**

The paper identifies two key challenges in VLM-based architectural learning: 1) existing building-age datasets are either geographically limited or temporally shallow, and 2) VLMs tend to memorize landmarks rather than truly learn architectural features. To address these challenges, the authors introduce YearGuessr, the first open benchmark for building-age estimation, containing 55,546 images from 157 countries spanning 1001–2024 CE. They also propose a baseline model, YearCLIP, which first classifies architectural styles and then computes the similarity score of a regressor to predict the precise year, achieving a balance between style recognition and temporal accuracy.

**Strengths:**

1. The paper clearly identifies the main challenges in training VLMs and introduces YearGuessr and YearCLIP, offering novel ideas for effective model training.

2. The figures and their accompanying explanations are clear, vivid, and concise.

**Weaknesses:**

1. Although the introduction highlights the current challenges of VLMs and the features of YearGuessr, the content could be more concrete. Additionally, the baseline model YearCLIP should be introduced in this section.

2. While the dataset covers many countries, there remains a Western-centric bias, which may limit the generalizability of the conclusions to other regions.

3. The description of the YearCLIP implementation in Section 4 is overly brief. The overall process illustrated in Figure 4 should be explained in more detail.

4. Focusing solely on building-age prediction makes the overall task relatively simple. Expanding the scope or content could make the paper more in-depth.

**Questions:**

Please address the weaknesses.

---

### Official Review · Reviewer_DUim · 2025-10-28

**Soundness:** 3
**Presentation:** 3
**Contribution:** 3
**Rating:** 8
**Confidence:** 3

**Summary:**

This paper introduces YearGuessr, a large-scale, open dataset for building age estimation, and proposes YearCLIP, a CLIP-based ordinal regression model with reasoning prompts and geolocation fusion. The dataset includes 55,546 Wikipedia-sourced façade images from 157 countries, labeled with construction years from 1001–2024 CE, GPS coordinates, and page-view statistics.

The paper investigates memorization bias in vision-language models (VLMs), showing that popular landmark buildings yield up to +34% higher accuracy for models like Gemini 2.0 compared to lesser-known structures — suggesting memorization rather than genuine architectural understanding.

Comprehensive experiments compare 30+ models, including CNNs, Transformers, CLIP-based, and both open and closed VLMs, under multiple metrics (MAE, interval accuracy, and popularity-aware MAE).

**Strengths:**

1. Addresses the underexplored problem of architectural age estimation, a unique, interpretable, and societally relevant visual reasoning task.

2. Provides a new lens on VLM memorization using an objective, measurable setup (year prediction) rather than subjective prompts.

3. Largest open dataset for building age estimation (55k images across 157 countries). Fully licensed under CC BY-SA 4.0 — ensuring reproducibility and openness. Includes rich metadata (captions, GPS, page views), enabling multi-modal and bias analysis.

4. Introduces ordinal regression + reasoning prompt integration within a CLIP backbone.

5. Adds zero-convolution fusion for coordinate conditioning.

6. Novel popularity-aware metric captures memorization bias.

7. Benchmarks across 43 models, spanning CNNs to latest VLMs (Gemini 2.0, Grok2, Claude 3).

**Weaknesses:**

1. Dataset heavily skewed toward Western and modern architecture (85% from Americas + Europe), which may limit claims about global generalization.

2. While performance gaps across popularity bins are compelling, the paper does not explicitly test for data overlap between Wikipedia and VLM pretraining corpora. Claims of “memorization” could be further substantiated via retrieval or content leakage analysis.

**Questions:**

None.

---

### Official Review · Reviewer_mqHK · 2025-10-29

**Soundness:** 2
**Presentation:** 3
**Contribution:** 2
**Rating:** 2
**Confidence:** 4

**Summary:**

This paper discusses the popularity bias for building age estimation. The paper (1) claims that existing vision-language models perform well on well-known landmarks due to memorization from pretraining data but fail on ordinary buildings, and (2) proposes a new open benchmark and a baseline model to expose and measure this bias through ordinal regression and popularity-aware evaluation. Experimental results using the YearGuessr dataset show the promising performance of YearCLIP. The paper is well-structured and clearly presented on its novelty and contribution highlights. The main contributions of this paper are the proposed YearGuessr, a benchmark revealing and quantifying popularity bias, and the proposed method YearCLIP to mitigate this bias.

**Strengths:**

S1. The paper sets up an open corpus with global coverage, large-scale ordinal labels, GPS, and rich textual descriptions for building-age estimation.

S2. It is observed that some VLMs recognize familiar images rather than reasoning from stylistic or material cues, which reflects a strong popularity bias.

S3. The paper proposes a reason-enhanced model based on NumCLIP, offering transparency into the model’s decision-making process.

**Weaknesses:**

W1. The paper claims in Section 3.3 that it reports both with- and without-GPS results, but that comparison is not actually reported. The paper should add the results and analyze whether models simply memorize the GPS coordinates.

W2. The proposed dataset may have label noise. Construction years are taken directly from Wikipedia without consistent verification, introducing potential inaccuracies. Some textual descriptions are processed via LLMs to infer renovation status, which introduces secondary uncertainty.

W3. The paper overuses the `\paragraph{}` command, applying it to nearly every paragraph, which disrupts the reading flow, making the paper feel fragmented. It should be replaced with normal paragraph spacing to improve readability.

W4. The authors correctly note the geographic and temporal bias, changing the task to predicting building age and location jointly without GPS coordinates may mitigate the bias and prevent the model from memorizing the GPS coordinates.

**Questions:**

The font size in Table 3 is slightly small. It might be split into two tables.

**Details Of Ethics Concerns:**

There are hidden texts behind the caption of Fig. 6.

---

### Official Review · Reviewer_K9hr · 2025-10-31

**Soundness:** 2
**Presentation:** 3
**Contribution:** 2
**Rating:** 2
**Confidence:** 4

**Summary:**

The paper releases YearGuessr, a 55,546-image benchmark of building facades with construction years, GPS and page-view metadata, collected from Wikipedia/Wikimedia via a crawl + dedup + CLIP(B/32, text: “a building facade”) filter + light manual audit pipeline. It further proposes YearCLIP, a CLIP-based baseline with ordinal regression and GPS fusion, and evaluates 40+ CNN/Transformer/CLIP/VLM models. The core claim is that VLMs achieve much higher IA5 on famous landmarks (e.g., Gemini2.0-Flash: +34.18% IA5 from low- to high-popularity bins), which the authors interpret as memorization rather than architectural understanding.

**Strengths:**

- Public resource: a large, globally scoped benchmark (55,546 images) with curated splits and basic documentation.
- Clear phenomenon: popularity-stratified IA5 reveals a strong VLM popularity gain (up to +34.18%), which is practically relevant to leakage/memorization discussions.
- Baseline clarity: YearCLIP and the evaluation protocol (MAE, IA±{5,100}, popularity gain) are described sufficiently to reproduce headline numbers.

**Weaknesses:**

**W1. CLIP-driven cleaning dominates selection and can structurally favor CLIP-family models at evaluation.**
- About data processing. According to Fig. 2 and the surrounding description, the raw crawl proceeds through deduplication (retaining one image per page title; –8,346 images), then a CLIP(B/32) similarity filter to the text “a building facade”, which drops –26,303 images in the figure (the main text nearby states –26,338), followed by a very small manual audit (–35) on the test split. In other words, 26,303 of the 34,684 total deletions (~75.8%) are decided by CLIP, while the final human pass is negligible by comparison. This makes the resulting test distribution strongly conditioned by CLIP’s implicit notion of “facade-ness.”
- Unfair evaluation. It is a widely shared understanding that data cleaning is the most difficult and labor-intensive phase of dataset construction; here, however, the heavy lifting is effectively delegated to a single model family (CLIP). Because the same family is then evaluated as part of the benchmark, the setup naturally invites a selection–evaluation coupling: a dataset curated predominantly by CLIP may privilege CLIP-like feature geometry and filtering biases. Under this coupling, it is not surprising if CLIP-family baselines outperform non-CLIP CNN/Transformer baselines—even when broader-capacity VLMs/MLLMs (e.g., Qwen25VL-3B/7B/32B, etc.) are expected to set a higher ceiling on visual reasoning in general. This is precisely the counter-intuitive situation we observe: despite the presence of strong VLMs in the tables (e.g., the Qwen25VL series), CLIP-based methods can appear comparatively advantaged on this benchmark in relative terms to non-CLIP vision baselines, which is difficult to interpret without decoupling selection from evaluation.

- Why this matters for method development and leaderboard incentives: If the benchmark’s retained set is implicitly aligned to CLIP’s facade prior, then CLIP-derived methods may achieve SOTA with comparatively modest innovations, whereas stronger general-purpose VLMs (e.g., next-gen multi-modal LLMs) might underperform relatively on this dataset unless they emulate CLIP’s specific inductive biases—an undesirable, benchmark-induced incentive. From an ecosystem perspective, this could steer future work toward CLIP-like pipelines simply because they fit the cleaning prior, not because they generalize better to non-Wikipedia, identifier-masked, or OOD buildings.
- I am open to raising my score if the authors can empirically demonstrate that CLIP-based cleaning and CLIP-family evaluation are decoupled—e.g., by passing two of the diagnostics above with stable CLIP vs. non-CLIP gaps and by showing no adversarial alignment to CLIP in the discarded-vs-retained attribute report. Given the magnitude of CLIP’s role in the cleaning step (≈75.8% of deletions; figure) and the minimal human audit (–35 items), the burden of proof lies with the authors to rule out family bias.


**W2. No systematic comparison to existing datasets; redundancy and regional bias remain unaddressed**
- Although the work positions YearGuessr as a global benchmark, the paper does not provide a detailed, dataset-level comparison to existing building/ facade/ architectural datasets beyond a high-level figure. There is no quantification of cross-dataset redundancy or overlap (image-level or source-level), no per-region replication analysis to establish novelty relative to well-covered local/continental corpora, and no articulation of complementarity where overlap is expected.
- The geographical distribution is visibly skewed, with certain regions dominating the sample mass; there is no evidence of targeted supplementation for underrepresented areas to mitigate this bias. Taken together, the absence of a granular side-by-side comparison and a redundancy audit, combined with the persistent regional skew, weakens the claim that this dataset constitutes a genuinely new, globally balanced resource rather than a re-aggregation centered on regions already well represented elsewhere.

**W3. Pipeline offers little new methodological insight; claims are inconsistent across sections.**
- The data pipeline is a standard crawl→dedup→CLIP filter→light audit with no novel acquisition or bias-control mechanism beyond off-the-shelf CLIP filtering; the proposed YearCLIP baseline is a straightforward composition (CLIP image encoder + ordinal regression + GPS fusion + predefined prompts) without a compelling new architectural idea. This reads as resource curation plus a conventional baseline rather than a methodological advance.
- The submission text in the OpenReview page states “ordinal training halves MAE,” whereas the main results section reports a 13.5% reduction vs. GeoCLIP (45.69 → 39.52), not a halving. Please unify the claim with the actual tabled results.

**W4. Evidence for “VLM memorization” is correlational; key controls are missing.**
- The popularity gain is reported, but there is no causal control separating landmark identification from style learning (no identifier-masking, no cross-source replication, no near-duplicate/leakage audit against pretraining). Please add: (i) an identifier-masked test (crop/blur landmark signage/skyline/text and re-test); (ii) a non-Wikipedia image source subset; (iii) a near-duplicate audit (perceptual hash / k-NN) against known pretraining collections; and (iv) CIs/p-values for popularity gains.

**Questions:**

Please refer to Weakness.

---

### Note · Authors · 2025-11-14

I have read and agree with the venue's withdrawal policy on behalf of myself and my co-authors.